# Environmental and ecological controls of the spatial distribution of microbial populations in aggregates

**Eloi Martinez-Rabert** [1] *, **Chiel van Amstel** [2], **Cindy Smith** [1], **William T. Sloan** [1], **Rebeca Gonzalez-Cabaleiro** [2]

**1** James Watt School of Engineering, Infrastructure and Environment Research Division, University of Glasgow, Advanced Research Centre, Glasgow, United Kingdom, **2** Department of Biotechnology, Delft University of Technology, Delft, Netherlands

* 2424069M@student.gla.ac.uk

**Data Availability Statement:** The source code and data used to produce the results and analyses

## Abstract

In microbial communities, the ecological interactions between species of different populations are responsible for the spatial distributions observed in aggregates (granules, biofilms or flocs). To explore the underlying mechanisms that control these processes, we have developed a mathematical modelling framework able to describe, label and quantify defined spatial structures that arise from microbial and environmental interactions in communities. An artificial system of three populations collaborating or competing in an aggregate is simulated using individual-based modelling under different environmental conditions. In this study, neutralism, competition, commensalism and concurrence of commensalism and competition have been considered. We were able to identify interspecific segregation of communities that appears in competitive environments (*columned stratification*), and a layered distribution of populations that emerges in commensal (*layered stratification*). When different ecological interactions were considered in the same aggregate, the resultant spatial distribution was identified as the one controlled by the most limiting substrate. A theoretical modulus was defined, with which we were able to quantify the effect of environmental conditions and ecological interactions to predict the most probable spatial distribution. The specific microbial patterns observed in our results allowed us to identify the optimal spatial organizations for bacteria to thrive when building a microbial community and how this permitted co-existence of populations at different growth rates. Our model reveals that although ecological relationships between different species dictate the distribution of bacteria, the environment controls the final spatial distribution of the community.

## Author summary

Microbial communities are assembled by the interactions between microorganisms and the local environment. To fully understand and control the formation of microbial aggregates, we need to unravel the principles of both cell-cell, cell-environment and cell-space interactions. Until now, most studies have focused predominantly on single interactions

presented in this manuscript are available on a public GitHub repository at https://github.com/Computational-Platform-IbM/IbM.

**Funding:** This study was funded by the University of Glasgow James Watt EPSRC Scholarship (EP/R513222/1) awarded to EMR. The funders had no role in study design, data collection and analysis, decision to publish, or preparation of the manuscript.

**Competing interests:** The authors declare no competing financial interests.

between two microbes. However, microbial ecology is more complex than that, and multiple ecological interactions contribute to microbial community assembly. The identification of distinct spatial distributions of bacteria is a first step towards the understanding the underlying biological mechanisms that govern aggregate formation. Here, we show that it is possible to evaluate the influence of multiple ecological interactions and the environment on microbial community assembly through mathematical modelling. We have been able to distinguish interspecific segregation of communities in competition, and layered distribution in commensalism. When we considered more than one ecological interaction between populations, the resultant spatial distribution was identified as the one controlled by the most limiting substrate. Additionally, we defined a theoretical modulus that able us to predict the most probable spatial distribution under specific environmental conditions.

## Introduction

Microorganisms are the most diverse and widespread forms of life on Earth [1–3] and key players of global biogeochemical cycles [4]. Any microorganism that is part of a community is influenced by their neighbouring cells, either from the same species (intra-species interactions) or different (inter-species interactions). Ecological interactions (eco-interactions) among microorganisms are classified according to the net effect on each of the two interacting species–positive impact (+), negative impact (–) or no impact (0) (Fig 1) [5,6].

In ecosystems, multiple eco-interactions between microorganisms are found combining both positive and negative interactions. For example, ammonia-oxidizers (AO) and nitrite-oxidizers (NO) collaborate on the oxidation of ammonia to nitrate (substrate-related commensalism) while simultaneously competing for oxygen [7].

In some cases, microorganisms generate aggregates either by attachment onto a solid surface (biofilm) or self-immobilization of microbes (floc/granule) [8–10]. The formation of microbial aggregates can be regarded as a multiple-step process, to which physiochemical

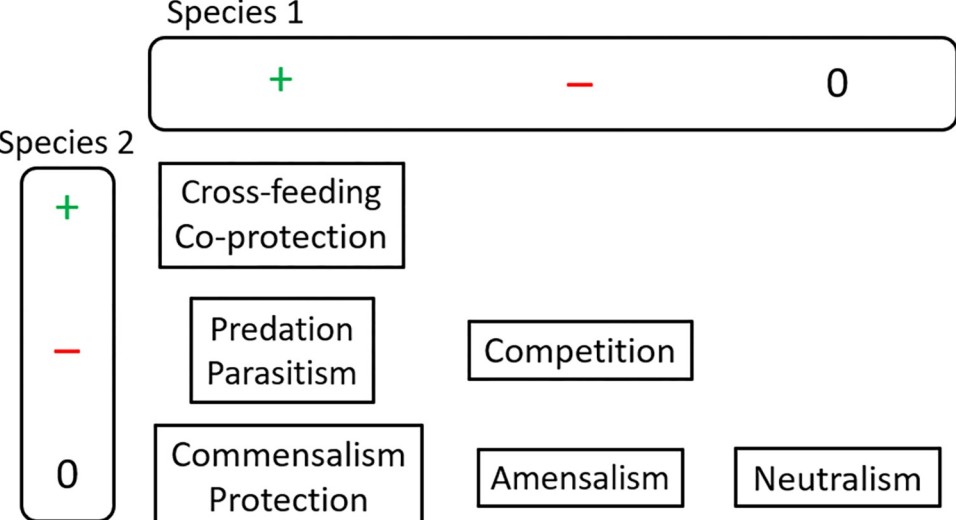

**Fig 1. Classification of eco-interactions between species.** Cross-feeding (or *syntrophy*) and co-protection (or *symprostasy*) are two specific eco-interactions belonging to mutualism (positive effect for both interacting species).

forces and biological properties play a crucial role [11,12]: (*1*) cell-to-cell contact, (*2*) attractive forces between cells causing them to aggregate, (*3*) microbial colonies formation and maturation of microbial aggregate, (*4*) establishment of the final three-dimensional structure of microbes shaped by shear forces and dispersion/invasion of planktonic microbes.

The process of aggregate formation is shaped by a reciprocal organism–environment influence called the "*dynamic fitness landscape*" [13,14]. However, ecological interactions (organism–organism influence) also play a fundamental role in microbial community assembly [15–17]. In this work we have adopted the *dynamic fitness landscape* concept but considering organism–organism influence, defining different *ecological environments* for specific eco-interactions between populations in community.

The mechanisms that control spatial distributions of microbial populations, the influence of the environmental factors to microbial community assembly (cell-environment interactions), and the ecological impact of the most relevant spatial distributions to microbial community (cell-space interactions) are still poorly understood. Visual analysis of spatial distribution of microbial populations help us to comprehend the function and the dynamics in aggregated systems [18]. The link between interspecies interactions and spatial organization of species has been already stablished for biofilm systems [18, 19]. Among all possible eco-interactions, cooperation/mutualism between bacterial species (or phenotypes) and the competition with cheaters are the most studied systems in this topic [19–23]. However neutralism, competition or commensalism are more common than cooperation/mutualism among cultivable bacteria [24]. Here, we use a mechanistic model to simulate the maturation process of microbial aggregates considering neutralism, competition and commensalism (S1 Appendix) to expand our comprehension of how cell-cell, cell-environment and cell-space interactions affect microbial community assembly in aggregates (relative abundances, microbial fitness, microbial colony formation and spatial distribution of microbial populations). Additionally, the concurrence of multiple eco-interactions was also evaluated (competition plus commensalism). Our results reveal how the ecological environment (established by the existing interactions among microbial communities and local conditions) controls the spatial organization of bacteria and the overall community assembly in aggregates.

## Results/Discussion

To explore the intrinsic influence of eco-interactions on the development of microbial communities growing in aggregates, a multispecies system was simulated using individual-based modelling (IbM). Examples in literature of empirical tests have demonstrated the ability of IbM to make accurate predictions for real biological systems that can offer mechanistic explanations of underlying biological processes [19,22,23,25–31].

Briefly, the model considers a two-dimensional space in which self-attached microorganisms of three different populations (identified as B1, B2 and B3) grow and divide. The three populations considered have the same characteristics and growth kinetics, therefore the community maintains a theoretical equal-fitness (see S1 Table). The theoretical equal-fitness assumption allowed us to elude the influence of growth kinetics upon the microbial community assembly and also to observe the genuine impact of the eco-interaction/s over the microbial fitness. The three populations have different metabolic stoichiometries, which, in turn, define the eco-interaction/s between them. The net effects of eco-interactions among the partners are presented here as symbols in a square bracket [19]. Following the series [B1,B2,B3]–neutralism [0,0,0]; competition [–,–,–]; commensalism [0,+,+] (Table 1). A constant concentration of substrates was fixed at the limits of the simulation domain (Dirichlet boundary conditions) which diffuse throughout the aggregate, generating local gradients in substrate

**Table 1. Definition of eco-interactions.** Summary of ecological net effect on bacteria and their respective metabolic stoichiometries.

| | | Ecological net effect | Metabolic stoichiometries [b] | | | |
|---|---|---|---|---|---|---|
| | | (0,– or +) [a] | A | B | C | D |
| **Neutralism** [0,0,0] | B1 | 0 | $-1/Y_{XS}$ | 0 | 0 | $1/Y_{XS}$ |
| | B2 | 0 | 0 | $-1/Y_{XS}$ | 0 | $1/Y_{XS}$ |
| | B3 | 0 | 0 | 0 | $-1/Y_{XS}$ | $1/Y_{XS}$ |
| **Competition** [–,–,–] | B1 | – | $-1/Y_{XS}$ | $1/Y_{XS}$ | 0 | 0 |
| | B2 | – | $-1/Y_{XS}$ | $1/Y_{XS}$ | 0 | 0 |
| | B3 | – | $-1/Y_{XS}$ | $1/Y_{XS}$ | 0 | 0 |
| **Commensalism** [0,+,+] | B1 | 0 | $-1/Y_{XS}$ | $1/Y_{XS}$ | 0 | 0 |
| | B2 | + | 0 | $-1/Y_{XS}$ | $1/Y_{XS}$ | 0 |
| | B3 | + | 0 | 0 | $-1/Y_{XS}$ | $1/Y_{XS}$ |

[a] Label legend: no effect (0), negative effect (–), positive effect (+).

[b] Units: (mol S)·(mol X)$^{-1}$. Negative value: consumption. Positive value: production.

concentrations. In all experiments, the aggregates grow until the relative abundance of the microbial populations, substrate/product concentrations in the bulk liquid, and actual growth rate ($\mu$) remain unchanged therefore reaching steady state (unless otherwise indicated in the caption of the figure). In these conditions microbial fitness (F) can be calculated as the actual microbial growth rate ($\mu$), because a single environment was imposed in our simulations [32]. We did not consider active movement nor dispersion/invasion, but microbes could move passively due to shoving forces exerted by neighbouring individuals as they grow and divide (see Methods).

## Influence of single eco-interaction on microbial communities

Three single eco-interactions were defined: neutralism (no interaction between populations), competition (all microbial populations consume the same resource), and commensalism (some microbial populations consume the metabolic product yielded by others). In a first step, independent simulations were designed with the objective to identify the formation patterns of spatial structures that are associated to each ecological interaction under different total substrate concentrations ($[S]_T$, see S3 Table).

**Neutral environment and the inevitable competition for space.** Mitri et al. (2016) [29] concluded that spatial organization of microbial communities was only observed when resources were limited. Under conditions of excess of substrates, the microbial colonies remained well-mixed (i.e., high colony heterogeneity). From an ecological viewpoint, resource availability has a direct impact on interactions among species. For example, a limited resource environment intensifies the competition for the substrate. Then, the influence of the eco-interactions on community assembly could be modified by changing the resource availability up to a hypothetic null effect. A set of simulations considering the three different eco-interactions defined (neutralism, competition, and commensalism), under different substrate concentrations (100 mM, 10 mM, 1.0 mM) were performed.

Fig 2 shows that our simulation results indicate that substrate limitation is the cause for the generation of spatial patterns of microbial communities. When substrate is not limiting inside the aggregate (simulations with $[S]_T$ = 100 mM), no particular spatial arrangement of microbial populations was observed in any of the considered eco-interaction (Fig 2A). Diffusion through the aggregate defines the observed level of colony heterogeneity (S1 Fig) [29]. When the substrate concentration in the bulk liquid was reduced from 100 mM to 10 and 1 mM

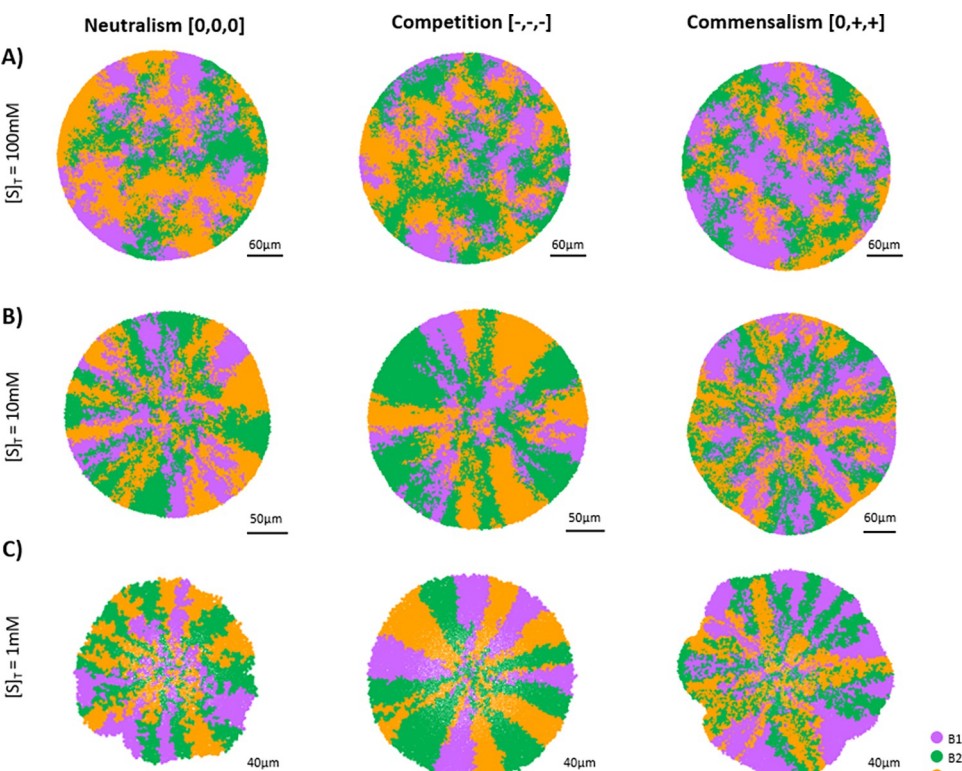

**Fig 2. Spatial distribution of microbial populations at different substrates concentrations.** (A) Aggregate pictures captured after 8 d of simulation considering neutralism, competition and commensalism with $[S]_T = 100$ mM. (B) Aggregate pictures captured at 10 d of simulation considering neutralism, competition and commensalism with $[S]_T = 10$ mM. (C) Aggregate pictures captured at 15 d of simulation considering neutralism, competition and commensalism with $[S]_T = 1$ mM. None of the simulations are in steady state yet. Black solid line on bottom-right of aggregates represent the scale bar. Substrate profiles on the transverse plane of aggregates have been included on S2 Fig.

(Fig 2B and 2C, respectively), distinctive spatial distributions of microbial populations emerged. Radial expansion of populations is associated to the presence of gradients of substrate concentrations (see S1 Video and S2 Video).

The well-mixed structures observed in this work and in Mitri *et al.* 2016 [29] are associated with environments in which substrates are not limiting (i.e., neutral environment), regardless the eco-interaction between microbial populations. However, in maturated aggregates with a certain size there will always be a gradient of concentrations of substrates [33,34].

In Fig 3, neutralism (null ecological interactions between populations) is analysed with more detail and in conditions of substrate limitation ($[S]_T \leq 1.0$ mM). In this situation, microbial populations consume different substrates, but the limitation of substrate availability increases the pressure for space competition between different populations. No significant differences between relative abundances of active bacteria were observed among populations (all around 33%, Fig 3A), but there was a significant difference between microbial fitness of populations at 1.0 mM (Fig 3B). In addition, a strong positive correlation between relative abundances and microbial fitness was observed at limiting substrate concentrations ($r = 0.760$–$0.947$, $p < 0.003$; S3A Fig).

In neutralism, some bacteria were able to remain active although they were farther from the source of nutrients (i.e., in the bulk liquid) than other species (Fig 3C). If the substrate is not consumed by the corresponding microbial population, mass transfer resistance (i.e., resistance to the net movement of substrates through the aggregate) is the only reason for substrate

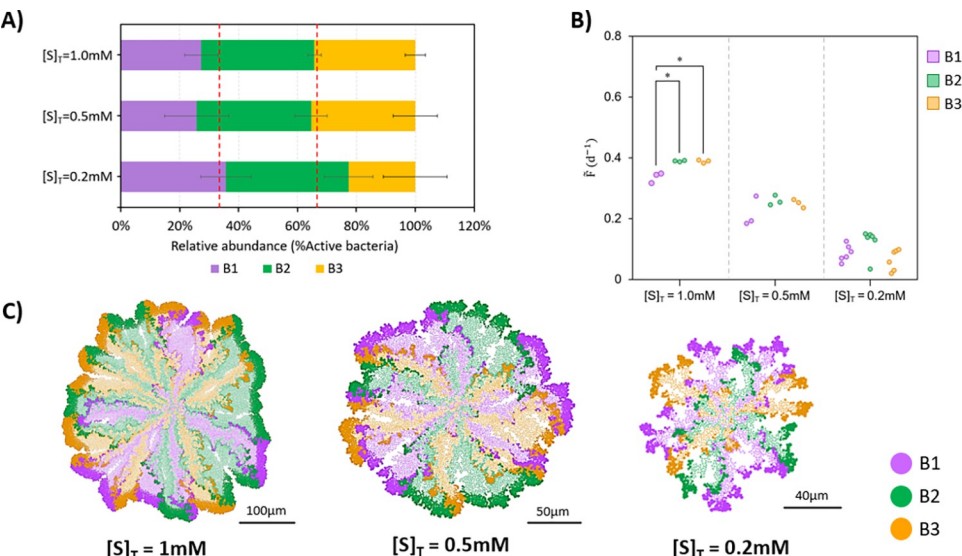

**Fig 3. Neutralism [0,0,0].** (A) Relative abundances of B1, B2 and B3 populations in the community. Dashed red lines indicate 33.33% and 66.66% relative abundances. (B) Microbial fitness (median, $\tilde{F}$) is calculated in all replicates ($n = 3$ for 1.0 mM and 0.5 mM, $n = 6$ for 0.2 mM). Asterisks indicate the significance level of the difference between B1, B2 and B3 specific growth rate. (C) Aggregates captured at steady state (75 d for 1.0 mM and 0.5 mM, 100 d for 0.2 mM). Inactive bacteria are shown in a lighter colour. Black solid lines on bottom represent the scale bars. Significance level legend: ns, not significant; *, $p < 0.05$; **, $p < 0.01$; ***, $p < 0.001$.

gradients, which explains the aforementioned positive correlation between relative abundances of active bacteria and microbial fitness. If more bacteria of certain species are on the external part of the aggregate, they would grow faster (increasing the fitness median) and, as consequence, be relatively more abundant. In contrast, those bacteria which are in a deeper position, have less available substrate due to the gradients formed and, consequently, grow slower. The aforementioned observations highlight the importance of the spatial distributions in aggregates and therefore the inherently present competition for space when substrates are limiting (cell-space interactions).

**Competition for substrate.** We next ask how the presence of competition for the same substrate could influence the spatial structure of the microbial community (Fig 4). We employed the same microbial community of three populations, but now competing for same substrate ($[S]_T \leq 1.0$ mM).

No significant differences on relative abundances or microbial fitness of populations were observed in the community (Fig 4A and 4B). In addition, no correlation between relative abundances and microbial fitness was found ($r = -0.040–0.290$, $p > 0.240$; S3B Fig).

Fig 4C shows radial distribution of microbial populations (*columned stratification*) at already early stages of the maturation of the microbial aggregate. Patterns of *columned stratification* on microbial populations in competitive environments is consistent with previous findings [19,29,35]. The similarity on microbial fitness observed in Fig 4B was maintained thanks to the spatial distribution that enabled all bacteria, whatever their relative abundances were, to get access to substrate. This differs with the previous results obtained when simulating neutralism where positive correlations between relative abundances and microbial fitness were observed (Fig 3C). These differences can be explained due to an increased substrate limitation in the competitive environment, which leads to larger substrate gradients and therefore to more defined radial spatial structures.

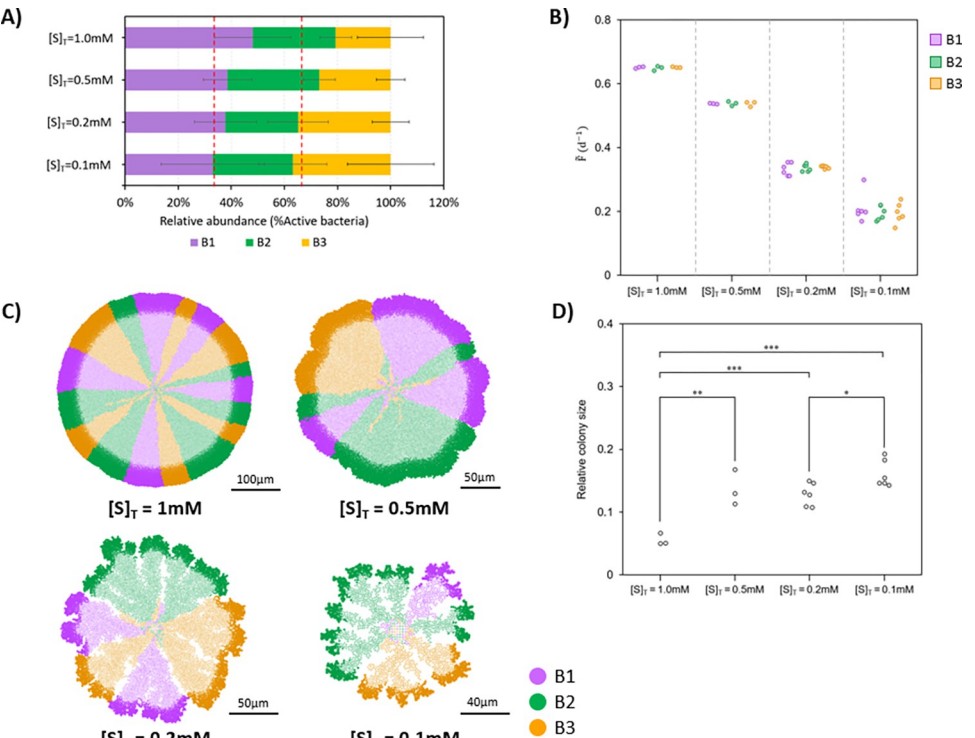

**Fig 4. Competition [−,−,−].** (A) Relative abundances of B1, B2 and B3 populations in the community. Dashed red lines indicate 33.33% and 66.66% relative abundances. (B) Microbial fitness (median, $\tilde{F}$) is calculated in all replicates ($n$ = 3 for 1.0 mM and 0.5 mM; $n$ = 6 for 0.2 mM and 0.1 mM). Asterisks indicate the significance level of the difference between the specific growth rate of B1, B2 and B3 populations. (C) Aggregates captured at steady state (50 d for 1.0 mM and 0.5 mM, 75 d for 0.2 mM and 0.1 mM). Inactive bacteria are shown in a lighter colour. Black solid lines on bottom represent the scale bars. (D) Average relative size of colonies (with respect to the total perimeter of the aggregate) is calculated in all replicates ($n$ = 3 for 1.0 mM and 0.5 mM; $n$ = 6 for 0.2 mM and 0.1 mM). Asterisks indicate the significance level of the difference between substrates concentrations. Significance level legend: ns, not significant; *, $p < 0.05$; **, $p < 0.01$; ***, $p < 0.001$.

We observed a negative correlation between substrate concentration and the relative colony size for the different populations (computed as the perimeter of a circular section that a population occupies without the interference of other populations in the community over the total perimeter of the aggregate, see Methods) (Fig 4D). When substrate availability is reduced, the competition for same resource (and for space) is intensified. The minimum colony size to thrive in a competitive environment will be higher as limitation for substrate increases.

**Substrate-related commensalism: division of labour.** A new set of simulations was performed considering substrate-related commensalism, in which population B1 feeds B2 and then, B2 feeds B3. From this interaction, populations B2 and B3 benefit from the community while B1 is neither benefited nor harmed: [0,+,+]. As in previous simulations, we forced substrate limitation inside the aggregate ([S]$_T$ ≤ 1.0 mM) to study the influence of the commensal environment on the microbial community assembly (Fig 5).

No significant differences in the relative abundance of bacteria between the simulations at different substrate concentrations were observed except at 0.5 mM and 0.1 mM (Fig 5A). Although the same growth kinetics are considered for all populations ($\mu_{max}$, $K_S$, $b_{max}$ and $Y_{XS}$), microbial fitness of B1, B2 and B3 populations were significantly different at the end of the simulations (Fig 5B). In all substrate concentrations, microbial fitness values followed the same pattern–B1 population grew faster than B2, and B2 grew faster than B3. This can be

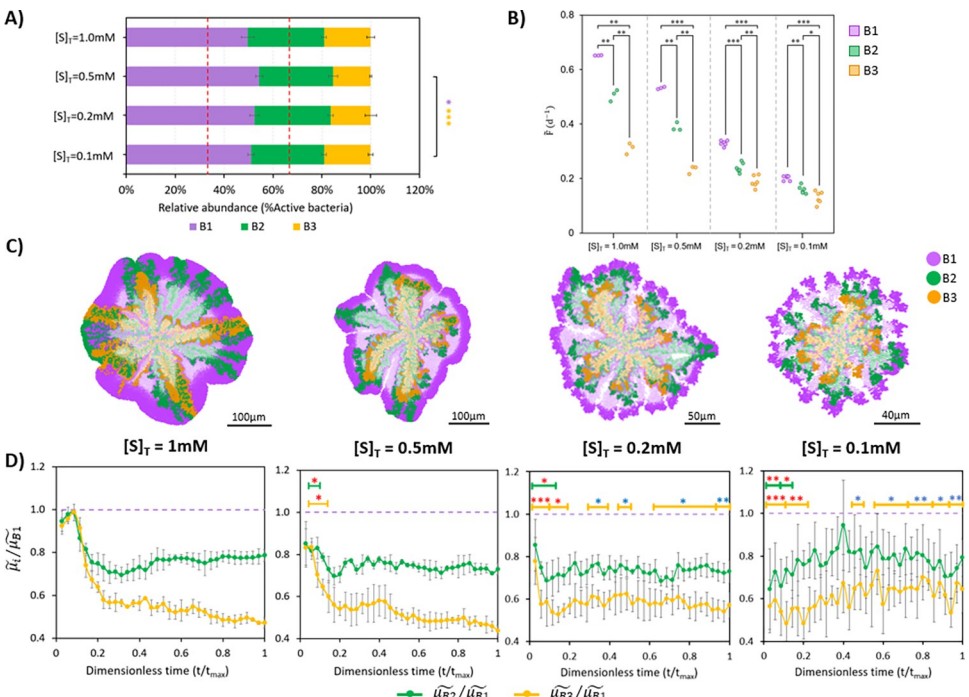

**Fig 5. Commensalism [0,+,+].** (A) Relative abundances of B1, B2 and B3 populations in the community. Dashed red lines indicate 33.33% and 66.66% relative abundances. Square brackets with asterisks indicate the significance level of the difference between relative abundances between populations. Colour of asterisks points out what bacteria it refers to. (B) Microbial fitness (median, $\tilde{F}$) is calculated in all replicates ($n = 3$ for 1.0 mM and 0.5 mM; $n = 6$ for 0.2 mM and 0.1 mM). (C) Aggregates captured at steady state (50 d for 1.0 mM and 0.5 mM, 75 d for 0.2 mM and 0.1 mM). Inactive bacteria are shown in a lighter colour. Black solid lines on bottom represent the scale bars. (D) Transient change of growth rate ratios of active B2 and B3 over B1 ($\tilde{\mu}_{B2}/\tilde{\mu}_{B1}$ and $\tilde{\mu}_{B3}/\tilde{\mu}_{B1}$, respectively) in all replicates ($n = 3$ for 1.0 mM and 0.5 mM; $n = 6$ for 0.2 mM and 0.1 mM). Growth rate ratios, normalise the comparison and reduce the influence of substrate concentration in the analysis. Dimensionless time was applied to compare simulations with different time length ($t_{max}$). Asterisks indicate the significance level of difference between 1.0 mM and the other concentrations (1.0 mM vs 0.5 mM; 1.0 mM vs 0.2 mM; 1.0 mM vs 0.1 mM). Colour of asterisks: red–growth rate ratio at [S] is lower than that at 1.0 mM; blue–growth rate ratio at [S] is higher than that at 1.0 mM. Significance level legend: ns, not significant; *, $p < 0.05$; **, $p < 0.01$; ***, $p < 0.001$. Width of the lines indicate the time to which the significance level corresponds. Colour of the lines: green–B2; orange–B3.

attributed to the commensal interaction and feeding regime. A strong positive correlation between relative abundances and microbial fitness was observed for all the substrate concentrations tested ($r = 0.950$–$0.982$, $p < 0.001$; S3C Fig).

Distinct spatial distributions emerged on the different limiting substrate concentrations (Fig 5C). At substrate concentrations of 1.0 mM, microorganisms which perform the sequential metabolic steps (B2 and B3) were found on the peripheral part together with their metabolic predecessor (B1). In simulations with 0.5 mM, only B1 and B2 were on the peripheral part and B3 remained always below their metabolic predecessors. Then, under stronger substrate limitations (0.2 mM and 0.1 mM) microbial communities generated a concentric disposition of active bacteria from different populations following the metabolic sequence of commensalism (*layered stratification*).

The transient change of the growth rate ratios of active B2 and B3 over B1 ($\tilde{\mu}_{B2}/\tilde{\mu}_{B1}$ and $\tilde{\mu}_{B3}/\tilde{\mu}_{B1}$, respectively) shows the contribution of space competition on microbial community assembly and the importance of the early stages of the maturation process (Fig 5D). Metabolic successors (B2 and B3) will always be able to stay at the peripheral zone (even if they grow slower than their predecessor B1) as long as they occupied enough circular region at the early

stages of maturation process. This happened when the growth rate of all the populations was similar at the beginning of maturation process. *Layered stratification* minimizes mass transfer resistances and favours the growth rate of the metabolic successors (Fig 5D) as the populations grow in the positions where higher concentrations of substrates are generated (S4 Fig).

The association of *layered stratification* with commensalism is observed in many environmentally relevant microbial processes (such as nitrification, organic anaerobic digestion or herbicide degradation) [36–39] and *in vitro* communities [19, 40, 41]. Momeni *et al.* (2013) [19] cultured two engineered yeast strains in which the metabolic predecessor took lysine from the media and overproduced adenine to feed the metabolic successor observing (and also predicting by modelling) *layered stratification*. The feeding regime in this study was different leading to lysine depletion, therefore populations were organised in a different order. In our case, the substrate of the metabolic predecessor was constantly fed on the system favouring its dominance.

## Concurrence of multiple eco-interactions

Microbial ecology is more complex than a single eco-interaction system. In the following simulations, the concurrence of two eco-interactions was considered, keeping the same microbial system of three populations: substrate-related commensalism (substrate A) and competition for substrate ($O_2$).

The results are presented in Fig 6 at different concentrations of substrate A (1.0 mM to 0.05 mM) and $O_2$ (10 mg/L to 1 mg/L). A *layered stratification* of microbial communities (associated with commensalism) was observed when concentration of substrate A was equal or lower than 0.1 mM and concentration of $O_2$ was equal or higher than 6 mg/L. *Columned stratification* of microbial communities (associated to competition of populations) was observed when

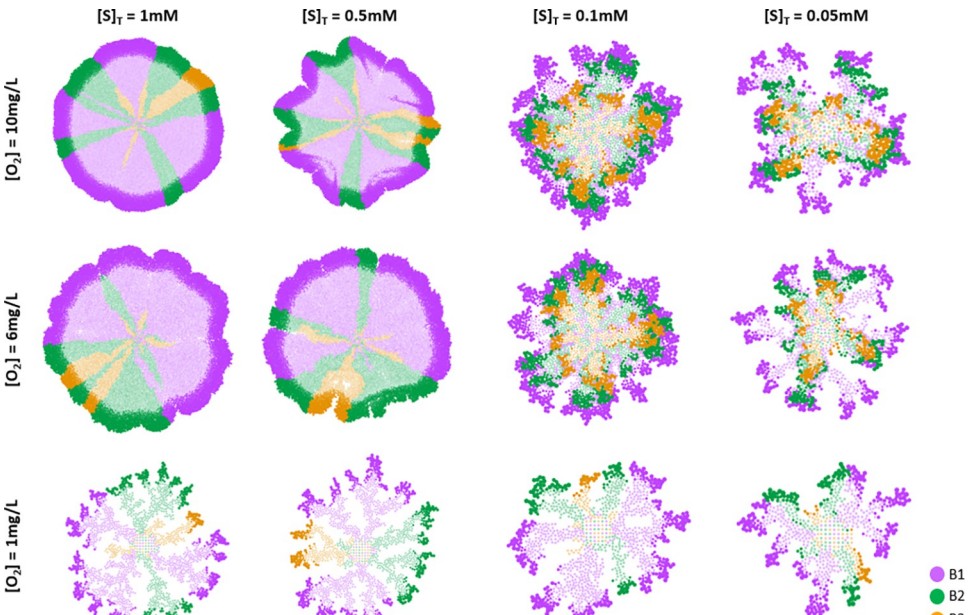

**Fig 6. Aggregates captured at steady state considering commensalism and competition for $O_2$.** Steady state times: 50d for $[S]_T$ = 1.0 mM, 0.5 mM, 0.1 mM and $[O]_2$ = 10 mg/L, 6 mg/L; 75d for $[S]_T$ = 1.0 mM, 0.5 mM, 0.1 mM and $[O]_2$ = 1 mg/L; 100d for $[S]_T$ = 0.05 mM and $[O]_2$ = 10 mg/L, 6 mg/L, 1 mg/L. Inactive bacteria are shown in a lighter colour. Substrate profiles on the transverse plane of aggregates have been included on S5 Fig. Aggregates captured at steady state.

the concentration of $O_2$ was 1 mg/L and/or concentration of substrate A was higher than 0.1 mM. The influence of the environment over the spatial distribution of microbial populations was also indicated by Momeni *et al.* (2013) [22], as the presence or absence of key substrates for cooperation (adenine and lysine) affected spatial distribution of cooperators.

Comparing the substrate profiles on the transverse plane that correspond to the spatial distributions presented in Fig 6 (see S5 Fig), a clear relationship between the substrate limiting and the observed final microbial spatial distribution was found. When substrate A was more limited than $O_2$ (commensal environment), *layered stratification* of populations was observed. In contrast, a *columned stratification* emerged when $O_2$ was more limited than substrate A (competitive environment). The limiting substrate establishes the ecological environment of the aggregate and therefore, the spatial distribution of microbial populations.

Which ecological environment controls the aggregated system (commensal or competitive environment) did not only influence the spatial distribution of microbial populations, but also the relative abundance of active bacteria (S6 Fig). When a commensal environment was present ($[S]_T \leq 0.1$ mM and $[O_2] \geq 6$ mg/L), a similar proportion of bacteria (B1:B2:B3) to single-interaction commensalism simulations (Fig 5A) was observed (1:0.6:0.3 for single-interaction commensalism; 1:0.6:0.4 for commensalism plus competition; $p > 0.05$). In contrast, when a competitive environment was present ($[S]_T > 0.1$ mM and/or $[O_2] = 1$ mg/L), a completely different proportion of bacteria to single-interaction competition (Fig 4A) was found (1:1:1 for single-interaction competition; 1:0.4:0.1 for commensalism plus competition; $p < 0.0001$). In a competitive environment, lower proportion of B2 and B3 populations than in commensal environment was observed. No clear trend of microbial fitness was found as result of the ecological environment (S7 Fig). In general, the predominant trend in microbial fitness was the same as in single-interaction commensalism ($F_{B1} > F_{B2} > F_{B3}$, Fig 5B), although in some competitive environments (e.g., 1 mM:6 mg/L, 1 mM:1 mg/L or 0.5 mM:1 mg/L; $[S]_T:[O_2]$) the fitness of the populations was not significantly different (like in single-interaction competition, Fig 4B). This can explain why a different proportion of B1, B2 and B3 was found when a competitive environment dominated (S8 Fig).

From an ecological perspective, the *columned stratification* found in competitive environments allows populations with lower fitness than their competitors to co-exist. This particular ecological influence of the spatial distribution is fundamental for the dominance of cooperation over cheating [22]. Therefore, established *columned stratification* might hinder the repression of certain microbial populations living in aggregated systems. However, less competitive species in aggregates must be able to compete for its space and survive at the early stages of maturation when the stratification is not formed yet. For example, if lone cooperator cells (less competitive due to investment in secretion) meet with the cheater before they have a chance to establish the colony, this cooperation will be inhibited by competition [21].

L*ayered stratification* was identified as the optimal spatial organization of microbial communities in which division of labour occurs (substrate-related commensalism). This microbial distribution is also observed in protective environments, in which the peripheral population would act as a protector of the others [42–44].

**Eco-interaction modulus ($\phi_{EI}$).** Concentration gradients of substrates in microbial aggregates are the result of mass transfer limitations and biological activity. Inevitably one of them will be the rate-limiting process depending on the conditions and the biological system. Substrate gradients only emerge when mass transfer resistance is sufficient to limit the rate of the reaction. In the presence of multiple substrates, the limiting one can be identified by comparing diffusion and reaction rates. We have defined the eco-interaction modulus ($\phi_{EI}$; Eq 1) in

which the biological Thiele modulus of involved substrates is compared (see Methods).

$$\phi_{EI} = \frac{\phi_A}{\phi_{O2}} = \frac{\sqrt{(n_A \cdot q_A)/(D_A \cdot C_A)}}{\sqrt{(n_{O2} \cdot q_{O2})/(D_{O2} \cdot C_{O_2})}} \tag{1}$$

Where $q_i$ is the specific substrate uptake rate, $D_i$ is the diffusion coefficient, $C_i$ is the concentration of substrate $i$ in bulk liquid, and $n_i$ is the relative abundance of the microbial population that consume the substrate $i$. If $\phi_{EI}$ is higher than 1.0, substrate A is the most limiting and therefore, a commensal environment would be present. If $\phi_{EI}$ is lower than 1.0, $O_2$ is the most limiting inside the aggregate and a competitive environment would be present.

The eco-interaction modulus ($\phi_{EI}$) presented in Eq 1 was defined for all simulation experiments considering commensalism (initiated by consumption of substrate A) and competition for $O_2$ under different concentrations (Fig 7). When $\phi_{EI}$ value was higher than 1.0, microbial communities were distributed following a *layered stratification* (commensal environment), whereas when $\phi_{EI}$ value was lower than 1.0, microbial communities were organized following a *columned stratification* (competitive environment). A particular distribution of microbial species was noticed when [A] and [$O_2$] was 0.1 mM and 3.75 mg/L, respectively ($\phi_{EI}$ = 1.07). In this case, a mixture of *layered* and *columned stratification* was observed, because the substrate limitation in the spatial distribution of populations was not uniform. In certain zones, $O_2$ was the most limiting substrate, in others the substrate A (see S9 Fig).

In addition to the environment (in this case, established by the substrate concentrations), the diffusivity of substrates through the aggregate plays a fundamental role on the control of the spatial distribution of microbial populations, especially for substrates with higher molecular weight [45, 46]. A sensitivity analysis shows how diffusivity can directly impact on community assembly S10 Fig.

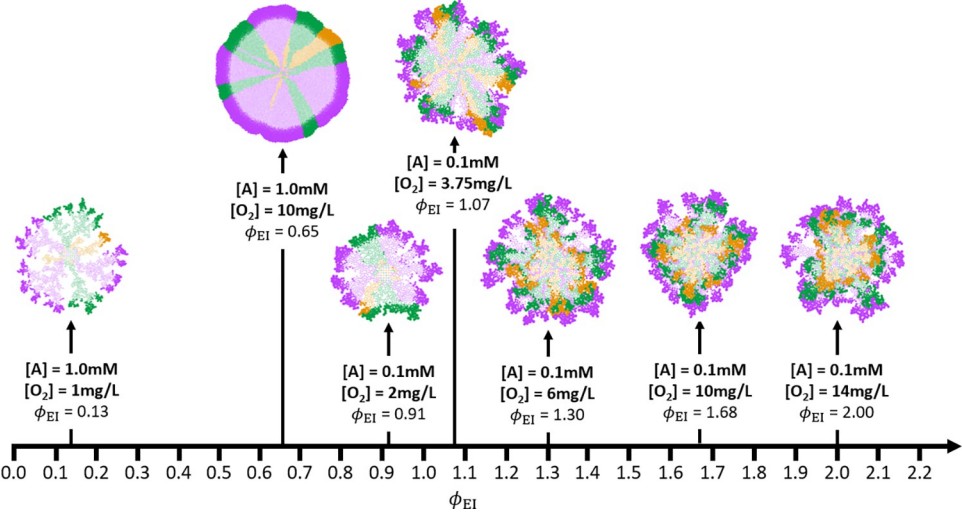

**Fig 7. Eco-interaction modulus diagram for simulations considering commensalism and competition for $O_2$ with [A] = 1.0–0.1 mM and [$O_2$] = 14–1 mg/L.** Aggregates captured at steady state (75 d for [S]$_T$ = 1.0 mM and [$O_2$] = 1 mg/L; 50 d for all other conditions). Inactive bacteria are shown in a lighter colour. Eco-interaction modulus calculated by Eq 1 for each replicate. Colour legend: B1 –purple; B2 –green; B3 –orange.

## Shear forces (detachment) controlling the spatial distribution of populations

We have described how the environment (substrate(s) concentration) together with the eco-interactions between species determine the spatial distribution of microbial populations. However, other environmental factors are also participants of the microbial community assembly in aggregates.

Shear forces, responsible for detachment in microbial aggregates, are one of the major factors involved in the formation of biofilms and granules. Steady state structures of aggregates are highly dependent on the shear forces, stablishing their thickness and density by microbial detachment [12]. Suarez *et al.* (2019) demonstrated that biofilm thickness has a significant impact on microbial community composition and spatial distribution [47]–a clear stratification of populations was observed in thick biofilms (400 μm), but not in thin ones (50 μm). In order to evaluate the influence of shear forces on the identified spatial distributions (*columned* and *layered stratification*), we simulated some of the conditions studied but now setting a maximum radius of aggregate of 30 μm. All bacteria that were more than 30 μm away from the centre of the granule were removed from the system (detachment) (Fig 8). Like in the study by Suarez *et al.*, the spatial distribution of microbial communities was lost when the maximum

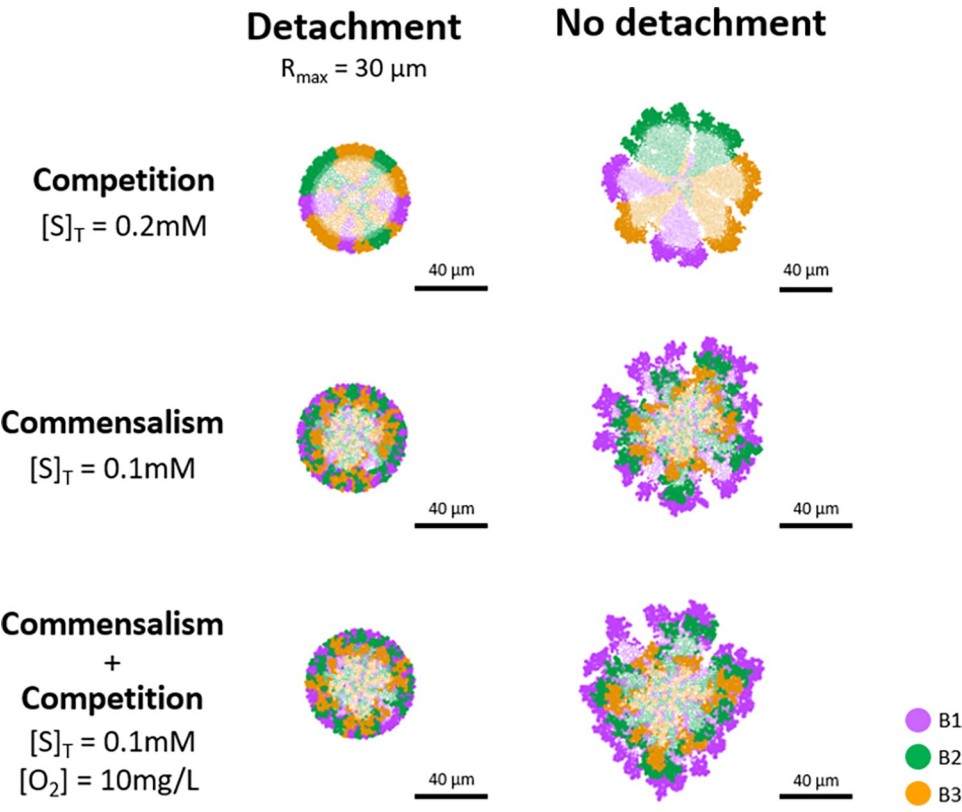

**Fig 8. Influence of shear force (detachment) to spatial distribution of microbial populations.** A maximum radius of 30 μm was set when detachment was considered. Aggregates captured at 50 d. Three conditions were evaluated–competition ($[S]_T = 0.2$ mM), commensalism ($[S]_T = 0.1$ mM), and commensalism + competition ($[S]_T = 0.1$ mM and $[O_2] = 10$ mg/L). Inactive bacteria are shown in a lighter colour. Substrate profiles on the transverse plane of aggregates have been included on S11 Fig. The simulation experiments including detachment were started with the same inoculum as those without detachment with the objective to observe the genuine impact of the shear force in the spatial distribution of the microbial populations.

radius was fixed for simulations involving commensalism or commensalism + competition (*layered stratification*), but not in competition (*columned stratification*) (Fig 8). When a maximum size of aggregate was applied, *layered distribution* was not observed because the available region that generated the specific microbial distribution was reduced significantly, and the metabolic successors (B2 and B3) were able to occupy the outer space of the aggregate that the metabolic predecessors (B1) left free once detached (S3 Video).

Our findings predict that *columned stratification* is more robust than *layered stratification* regarding detachment. The fragility of *layered stratification* might be a challenge for specific bioprocesses in which efficiency relies on generating this particular spatial distribution of microbial populations. Example of this can be the combination of partial nitrification and Anammox process in one stage [48].

## Conclusions

When analysing microbial growth in communities, the competition for space is generally overlooked. The results presented in this study show that competition for space influences the assembly of microbial communities in aggregates. We conclude that this competition for space implies that (*i*) neutral environment, without any particular distribution of populations, is only a transient state, (*ii*) in competition, the availability of space controls the colony size of the populations, and (*iii*) in commensalism their distribution. The spatial structures (controlled by ecological interactions) have in turn, implications on microbial growth and survival. The radial distribution of microbial populations (addressed here as *columned stratification*) increases the chance of less competitive individuals to thrive and to co-exist with populations that grow faster. On the other hand, the concentric disposition of communities (addressed here as *layered stratification*) would be the optimal distribution for metabolic division of labour (substrate-related commensalism). In addition, this study shows that although ecological relationships between different populations in aggregates dictate their distribution, the environment (operational conditions related to substrate concentrations, detachment and others) is controlling the observed final spatial distribution.

## Methods

### Description of the mathematical model

We have developed a multiscale model able to simulate the maturation of microbial communities growing in microbial aggregates. The starting point is a premature aggregate of 20 $\mu$m in which microbial species are randomly distributed. Briefly, the model is constituted by two submodels: (*i*) a physical model to simulate the diffusive transport of the dissolved substrates, (*ii*) a biological model that considers heterogeneity of the system and the intrinsic eco-interactions between microbial species.

The simulation domain is a two-dimensional and micro-scale space. In it, the diffusion of soluble components is resolved. These soluble components are the substrates and products of the microbial activity. The domain can be divided in three different zones: the aggregate, the boundary layer, and the bulk liquid (Fig 9A).

When bacteria grow and divide, they push each other, thereby increasing the radius of the overall aggregate. Diffusion of soluble components occurs throughout the aggregate where they are consumed or produced by the microorganisms. The boundary layer is the surrounding space of the aggregate defined to model the gradient of concentrations between the bulk liquid and the surface of the microbial aggregate. Only diffusion of the soluble components is resolved in this space. At the outside of the boundary layer, the gradient of concentration of all soluble species is considered negligible, assuming a well-mixed homogeneous reactor.

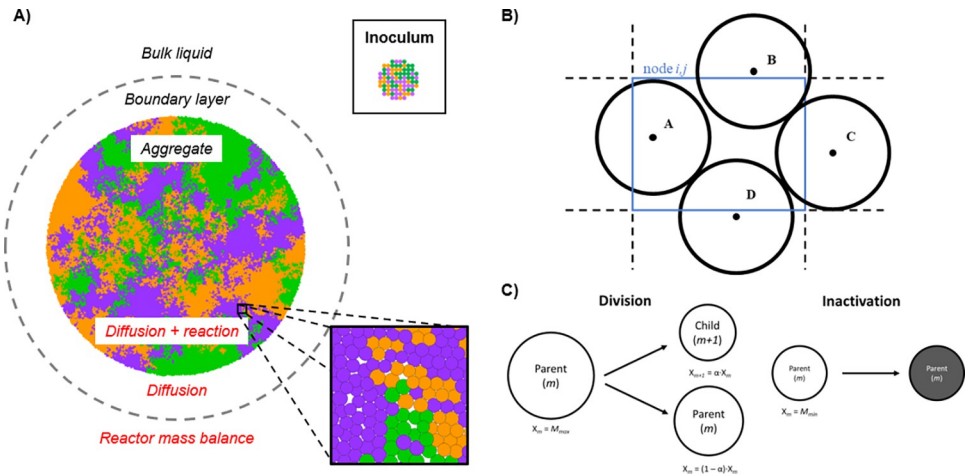

**Fig 9. Individual-based Model description.** (A) Representation of simulation domain. An expansion of specific aggregate zone (bottom-right square), and an example of the initial distribution of cells (top-right square) are shown, where microbes are represented as coloured circles with a specific radius: B1 –purple circle; B2 –green circle; B3 – orange circle. (B) Hypothetical position of microbes in the node. Only "microbe A" belongs to *node i,j* because its centre is inside the node. (C) Microbial division and inactivation. α is any stochastic value between 0.45–0.55. $M_{max}$ and $M_{min}$ refer to the maximum and minimum mass that microbe can reach, respectively.

**Diffusive transport model.** To describe the diffusion of components, the Fick´s second law equation is integrated over the time (*t*) and in space (*x* and *y*). The consumption and synthesis of the soluble components in this system is evaluated by each node of the simulation domain through the reaction term ($R(x, y, t)$; Eq 2). The reaction term is calculated according to the stoichiometry, the mass and growth rate of microbes present in the specific node. Additionally, steady sate for the production/degradation of soluble components was assumed. (see also *Reaction term* in S1 Appendix).

$$\frac{\partial}{\partial t}\phi(x, y, t) = \mathbb{D} \cdot \nabla_{xy}^2 \phi(x, y, t) + R(x, y, t) \tag{2}$$

Where $\phi(x, y, t)$ refers to the concentration of a soluble component in a position of the simulation domain (*x, y*) and in a time step (*t*), and $\mathbb{D}$ refers to the effective coefficient of diffusion. To simplify the resolution of Eq 2, a constant $\mathbb{D}$ value is considered for the whole aggregate [49] (see also *Discretization of diffusion-reaction equation* in S1 Appendix).

**Biological model: Individual-based Model (IbM).** The most distinguished properties of microbial aggregates are the presence of local gradients and the close localization of distinct microbial species. Our model aims to describes the heterogeneity of the aggregate and the intrinsic eco-interactions between microbial species. For this reason, each microorganism is modelled as a discrete entity with unique traits, a point of reaction in the spatial domain that shapes its kinetics as a function of the local environment. The position of microbe (and subsequent assignment of local conditions) is defined by the location of its centre (Fig 9B).

The growth of each individual is only affected by local conditions, which in turn are influenced by the activity of surrounding microbes. Like this, the model simulates the intrinsic interaction between the diverse microbes in the aggregate. Cellular division or death is assumed to occur at a certain microbe's size, independently of the state of other cells, only as a function of local conditions, capturing the heterogeneity of the aggregate and supporting non-linear growth solutions [50]. We did not consider cell lysis nor release of inter-cellular material (see stoichiometry of decay; S2 Table).

To define the actual growth for each individual ($\mu_m$, Eq 3), a specific stoichiometry is defined as well as Monod kinetic parameters–maximum specific growth rate ($\mu^{max}$), half-saturation constant ($K_S$), inhibition constant ($K_I$) and maintenance coefficient ($b^{max}$) [51]. See S1 and S2 Tables for kinetics and stoichiometries, respectively. The change of the mass of each specific individual is calculated by the differential equation Eq 4.

$$\mu_m^n = \mu_m^{max} \cdot \prod\left(\frac{\phi_{i,j}^n}{K_{S,m} + \phi_{i,j}^n}\right) \cdot \prod\left(\frac{K_{I,m}}{K_{I,m} + \phi_{i,j}^n}\right) - b_m \tag{3}$$

$$\frac{dX_m}{dt} = \mu_m^n \cdot X_m^n \tag{4}$$

Where $\phi_{i,j}^n$ refers to the concentration of substrate in node $i,j$ of the simulation domain in a time step $n$, and $X_m$ refers to the mass in moles of the cell $m$. The mass of each individual is integrated in time using a forward Euler scheme on Eq 4. Once the mass of each cell is known, its volume and radius ($r_m$) can be calculated defining a specific cell density ($\rho_m$) and assuming perfect spherical shape (Eq 5) [52].

$$r_m = \left(\frac{X_m}{\rho_m} \cdot \frac{3}{4\pi}\right)^{1/3} \tag{5}$$

The value of $\mu_m^n$ that Eq 3 returns, indicates if microbe $m$ is growing ($\mu_m > 0$) or dying ($\mu_m < 0$). The model assumes that once cells achieve a maximum mass ($M_{max}$), they divide. In this case, a new cell is formed (cell $m+1$, Eq 6) with an initial mass that is a random percentage ($\alpha$ is a stochastic parameter with a value between 0.45 and 0.55) of the total mass of the parent cell. The mass of the parent cell is updated with the mass remaining after the division (Eq 7).

$$X_{m+1} = \alpha \cdot X_m \tag{6}$$

$$X_m = (1 - \alpha) \cdot X_m \tag{7}$$

When $\mu_m < 0$ the individual shrinks, reducing its mass. When cells reach a minimum mass ($M_{min}$) or size considered negligible, they become inactive (i.e., they do not grow nor decay) and only can become active again if they increase their mass under more favourable conditions (Fig 9C).

When a cell divides, a random position for the new individual ($m+1$) in the neighbourhood of its parent ($m$) is assigned. When bacteria grow and/or divide a *shoving algorithm* checks the overlapping space between individuals. If this is bigger than the maximum overlap accepted, cells shove increasing the size of the aggregate (see *Shoving algorithm* in S1 Appendix).

**Integration.**   Cells are dividing in a time scale much slower than the diffusion-reaction process (~1 hour versus ~$10^{-8}$ hours). To solve the system, the model takes advantage of this time scale differentiation to separate the processes of solving the diffusion-reaction equation, and the cell division and its shoving [52].

First, the diffusion-reaction equation is integrated (with a time step $dt$) until it reaches a pseudo-steady state, in which the difference between the pseudo-steady state and the real steady state is less than a threshold (Eqs 8 and 9). To check whether pseudo-steady state is reached or not, only diffusion region is considered. This is because it is assumed that Dirichlet boundary condition and bulk liquid concentration are fixed in this time span, and only change when the microbial community change significantly (due to a cell division, cell inactivation or

a substantial variation in microbial mass).

$$[RES] = [L] * [\phi]^{\gamma} + (h^2/\mathbb{D}) \cdot R([\phi]) \tag{8}$$

$$Tol := max\left|\frac{RES_{i,j}}{1 \cdot 10^{-4} + \phi_{i,j}}\right| \leq 1\%, Tol := \begin{cases} max|RES_{i,j}| \leq 1\% \cdot \phi_{i,j}, if\ \phi_{i,j} \gg 1 \cdot 10^{-6} \\ max|RES_{i,j}| \leq 1 \cdot 10^{-6}, if\ \phi_{i,j} \ll 1 \cdot 10^{-6} \end{cases} \tag{9}$$

The integration continues updating each $n$-iterations the reaction term together with the integration of the microbial growth. When this pseudo-steady state is reached then, the mass balances of the overall reactor are integrated in a much bigger time step ($dt_{bac}$), function of the average microbial activity of the aggregate. Also, the biomass growth is integrated in this bigger time step. At the end of this bigger step, the Dirichlet boundary condition and the reaction term are updated. Therefore, the diffusion-reaction equation needs to be integrated again to reach a next pseudo-steady state. Each $n$ times that the diffusion-reaction equation reaches a pseudo-steady state ($dt_{div}$), the cell division is checked and if it happens, the algorithm of the microbial shoving is launched (see also *Integration* in S1 Appendix).

## Set of simulations

In all simulations, three different microbial populations have been considered named B1, B2 and B3. For each type of simulation experiment, the eco-interaction/s among microbial species is defined by their metabolic stoichiometries (neutralism, competition, commensalism or commensalism + competition; Fig 10) and substrate concentrations in the bulk liquid (range of concentrations). The simulations are run in triplicates or sextuplicate, using different random distribution of microbial species as starting. The details of simulation setups are given in S3 Table.

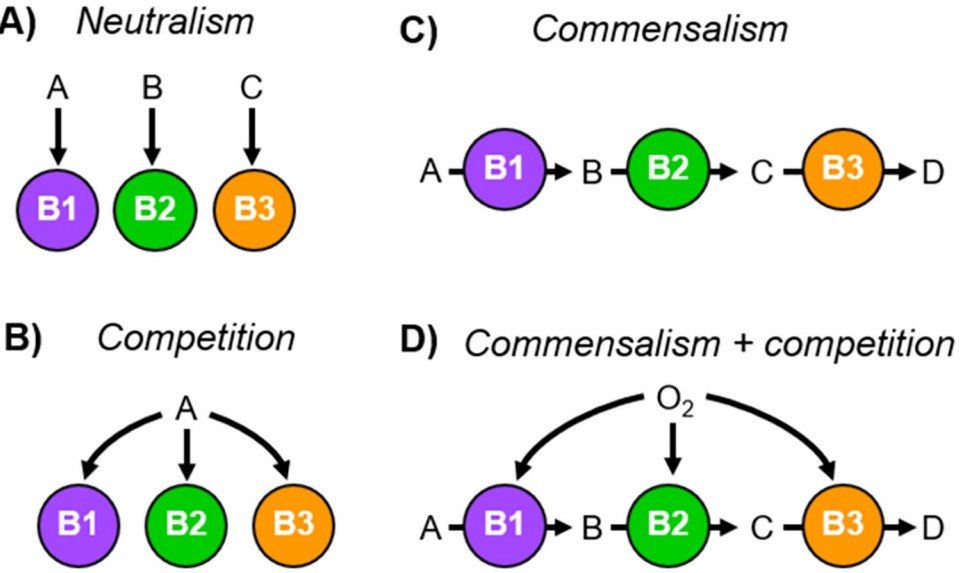

**Fig 10. Representation of eco-interactions among B1 (purple), B2 (green) and B3 (orange).** (A) Neutralism: [0,0,0]; no impact on B1, B2 and B3. (B) Competition: [–,–,–]; negative impact on B1, B2 and B3. (C) Commensalism: [0,+,+]; no impact on B1, positive impact on B2 and B3. (D) Commensalism + competition: [0,+,+] or [–,–,–]; impact on B1, B2 and B3 is defined by the environment.

## Calculation of colony size

Colony size ($P_c$) is the perimeter of circular section in which colony occupies, calculated by Eq 10.

$$P_c = 2\pi\langle r \rangle \left( \frac{\theta}{360} \right) \tag{10}$$

Where $\langle r \rangle$ is the average radius of colony section and $\theta$ is the angle of colony section. In this case, relative size of colonies was employed to neglect the influence of aggregate radius in our analysis (standardization of results). The relative colony size is defined as ($P_c / P_T$), where $P_T$ is the total perimeter of the aggregate.

## Definition of eco-interaction modulus ($\phi_{EI}$)

The Thiele modulus describes the relationship between the characteristic time for diffusion rate over the characteristic time for reaction rate, that is, between the surface reaction rate over the diffusion rate through the aggregate. Although Thiele modulus ($\phi$) was originally developed for immobilized catalysts [53], it has already been applied in microbial aggregates [54,55] to evaluate the influence of diffusional resistance in biological systems. The modified Thiele modulus for biological systems ($\phi_{Bio}$, Eq 11) was obtained by the ratio between characteristic time for diffusion ($\tau_{diff}$; Eq 12) and characteristic time for Monod-type reaction rate ($\tau_{r,Monod}$; Eq 13):

$$\phi_{Bio} = R \cdot \sqrt{\frac{q_S \cdot X}{D_S \cdot C_S}} \tag{11}$$

$$\tau_{diff} = \frac{R^2}{D_S} \tag{12}$$

$$\tau_{r,Monod} = \frac{Y_{XS}}{\mu_{max}} \cdot \frac{K_S + C_S}{X} \tag{13}$$

Where $R$ is the radius of aggregate (characteristic distance), $q_S$ is the specific uptake rate of substrate $S$, $X$ is the biomass concentration, $D_S$ is the diffusion coefficient of substrate $S$, $C_S$ is the concentration of substrate in bulk liquid, and $Y_{XS}$ is the growth yield coefficient. When $\phi_{Bio}$ value is large, internal diffusion of substrate limits the overall microbial activity. In contrast, when $\phi_{Bio}$ value is small, the biological reaction (uptake of substrates) is usually rate-limiting.

The next step is to correlate the eco-interactions that might influence the spatial distribution of microbial populations. In this case, a $\phi_{Bio}$ for substrate A (related to commensal environment) and another for $O_2$ (related to competitive interaction) ($\phi_A$ and $\phi_{O2}$, respectively) were defined. Then, the ratio of $\phi_A$ over $\phi_{O2}$ was performed obtaining the eco-interaction modulus ($\phi_{EI}$, Eq 14).

$$\phi_{EI} = \frac{\phi_A}{\phi_{O2}} = \frac{\sqrt{(n_A \cdot q_A)/(D_A \cdot C_A)}}{\sqrt{(n_{O2} \cdot q_{O2})/(D_{O2} \cdot C_{O_2})}} \tag{14}$$

Where $n_A$ is the relative abundance of microorganisms that consume A and $n_{O2}$ is the relative abundance of microorganisms that consume $O_2$.

### Statistical analyses

Statistical significance of the differences of relative abundances of B1, B2 and B3, and relative colony size among the tested substrate concentrations was assessed using the Welch's test. One population cannot be considered independent from the others in the same simulation, therefore, statistical significance between the microbial fitness of B1, B2 and B3 was assessed using the paired *t*-test. To evaluate the correlation between relative abundances of bacteria and microbial fitness (F), Person's correlation coefficient (*r*) was used.

## Supporting information

**S1 Appendix. Supporting Information.** Including the decision of the eco-interactions; explanation of discretization and methodology of diffusion-reaction equation; description of reaction term; details about shoving algorithm; and the algorithm of the integration process.
(PDF)

**S1 Table. Kinetics of all simulation setups.**
(PDF)

**S2 Table. Stoichiometry for all simulation setups.**
(PDF)

**S3 Table. Details of simulation experiments.** For each setup, eco-interactions between microbial species, environment of the reactor, and substrate concentrations on bulk liquid ([A], [B], [C] and [D]) are specified. Also, it includes in which Figures the results are shown.
(PDF)

**S1 Fig. Influence of inoculum size and substrate gradients on intermixing of microbial populations considering neutralism, competition and commensalism with $[S]_T = 100$ mM.** (A) Aggregate pictures captured at 8 d of simulation starting with an inoculum size of 20 μm (diameter) and considering diffusion resistance of substrates. (B) Aggregate pictures captured at 4 d of simulation starting with an inoculum size of 160 μm (replicating the starting point of Mitri *et al.* (2016)) [29] and removing the substrate gradients (no diffusion resistance). None of the simulations are in steady state yet.
(TIF)

**S2 Fig. Substrate profiles on the transverse plane of aggregates considering neutralism, competition or commensalism.** (A) Substrate profiles from simulations at $[S]_T = 100$ mM (t = 8 d). (B) Substrate profiles from simulations at $[S]_T = 10$ mM (t = 10 d). (C) Substrate profiles from simulations at $[S]_T = 1$ mM (t = 15d). Legend: [A]–purple line; [B]–green line; [C]–orange line.
(TIF)

**S3 Fig. Regression fits to different relative abundance/microbial fitness data pairs from simulation experiments with their corresponding Pearson's coefficient (*r*) with their significance value (*p*-value) and sample size (*n*).** (A) Simulations considering neutralism. (B) Simulations considering competition. (C) Simulations considering commensalism. Legend: B1– purple circles; B2 –green circles; B3 –orange circles.
(TIF)

**S4 Fig. Substrate profiles on the transverse plane of aggregates considering commensalism.** Legend: [A]–purple line; [B]–green line; [C]–orange line.
(TIF)

**S5 Fig. Substrate profiles on the transverse plane of aggregates considering commensalism and competition.** Legend: [A]–purple line; [B]–green line; [C]–orange line; [$O_2$]–black line.
(TIF)

**S6 Fig. Relative abundance of active bacteria from simulation experiments considering commensalism and competition.** Dashed red lines indicate 33.33% and 66.66% relative abundances. Colour of asterisks points out what bacteria it refers to. In the table are shown the significant level of the difference between B1, B2 and B3 relative abundances. Significance level legend: ns, not significant; *, $p < 0.05$; **, $p < 0.01$; ***, $p < 0.001$. Colours of y-axis text and table headers indicate the ecological environment (and spatial distribution of microbial populations) of simulation experiments: red–competitive environment (*columned stratification*); blue–commensal environment (*layered stratification*).
(TIF)

**S7 Fig. Microbial fitness (*F*) from simulation experiments considering commensalism and competition.** Asterisks indicate the significance level of the difference between B1, B2 and B3 specific growth rate. Significance level legend: ns, not significant; *, $p < 0.05$; **, $p < 0.01$; ***, $p < 0.001$.
(TIF)

**S8 Fig. Regression fits to different relative abundance/microbial fitness data pairs from simulation experiments considering commensalism and competition with their corresponding Pearson's coefficient (*r*) with their significance value (*p*-value) and sample size (*n*).** Legend: B1– purple circles; B2 –green circles; B3 –orange circles.
(TIF)

**S9 Fig. Ecological environment distribution in a hybrid stratification case considering concurrence of commensalism and competition ([S]$_T$ = 0.1 mM, [O$_2$] = 3.75 mg/L; ϕ$_{EI}$ = 1.07).** Inactive bacteria are shown in a lighter colour. The substrate profiles are from the transverse plane of aggregate.
(TIF)

**S10 Fig. Influence of substrate diffusivity on spatial distribution of microbial populations (commensalism + competition; [S]$_T$ = 1.0 mM, [O$_2$] = 10.0 mg/L).** Inactive bacteria are shown in a lighter colour. The substrate profiles (S11 Fig) are from the transverse plane of aggregates. Diffusion coefficient ($D_i$ in Eq 1) states the diffusion rate of certain substance into the fluid. Therefore, those substrates with lower diffusion coefficient will tend to be the limiting substrate in the aggregate, establishing the ecological environment and, consequently, the spatial distribution of microbial populations. In order to illustrate the influence of diffusion constant, we simulated again one of the cases of concurrence commensalism (substrate A) and competition (oxygen) but now reducing the diffusion coefficient of substrates A, B, C and D (from $3.60 \times 10^{-6}$ m$^2$/h to $0.5 \times 10^{-6}$ m$^2$/h). As example, the environment with 1.0 mM of A and 10.0 mg/L of O$_2$ (competitive environment, $\phi_{EI}$ = 0.65) was applied starting with the same inoculum. With the new diffusion coefficients, substrate A (instead of O$_2$) was the most limiting, obtaining a *layered stratification* of microbial populations (commensal environment, $\phi_{EI}$ = 1.51).
(TIF)

**S11 Fig. Substrate profiles on the transverse plane of aggregates considering or not detachment.**
(TIF)

**S1 Video. Development of aggregates and substrate profiles from simulations considering neutralism, competition and commensalism with $[S]_T$ = 10 mM (from 0 d to 10 d).** (AVI)

**S2 Video. Development of aggregates and substrate profiles from simulations considering neutralism, competition and commensalism with $[S]_T$ = 1 mM (from 0 d to 15 d).** (AVI)

**S3 Video. Development of aggregates and substrate profiles from simulations considering or not the influence of shear forces (detachment) (from 0 d to 50 d).** The simulation experiments including detachment were started with the same inoculum as those without detachment with the objective to observe the genuine impact of the shear force in the spatial distribution of the microbial populations. (AVI)

## Author Contributions

**Conceptualization:** Eloi Martinez-Rabert, Rebeca Gonzalez-Cabaleiro.

**Formal analysis:** Eloi Martinez-Rabert.

**Funding acquisition:** Cindy Smith, William T. Sloan.

**Investigation:** Eloi Martinez-Rabert.

**Methodology:** Eloi Martinez-Rabert, Rebeca Gonzalez-Cabaleiro.

**Software:** Eloi Martinez-Rabert, Chiel van Amstel.

**Supervision:** Rebeca Gonzalez-Cabaleiro.

**Validation:** Eloi Martinez-Rabert.

**Visualization:** Eloi Martinez-Rabert.

**Writing – original draft:** Eloi Martinez-Rabert.

**Writing – review & editing:** Eloi Martinez-Rabert, Chiel van Amstel, Cindy Smith, William T. Sloan, Rebeca Gonzalez-Cabaleiro.

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
