## [Decision Letter · Decision Letter 0]

1 Sep 2022

Dear Dr. Martinez-Rabert,

Thank you very much for submitting your manuscript "Environmental and ecological controls of the spatial distribution of microbial populations in aggregates" for consideration at PLOS Computational Biology.

As with all papers reviewed by the journal, your manuscript was reviewed by members of the editorial board and by several independent reviewers. In light of the reviews (below this email), we would like to invite the resubmission of a significantly-revised version that takes into account the reviewers' comments.

Although the reviewers are generally positive about the work, there are also concerns about the novelty of the work. Thus the advance over the previous modeling work should be clarified. I recommend including a more in-depth review of the existing work and to highlight the novel insights that are gained in the present work.  Also reviewers indicate the implications for the interpretation of in vitro results could be discussed in more detail as well as the comparison with and implications for real-life situations. Furthermore the reviewers ask why 3 out of a total of 6 possible competitive interactions have been studied - ideally the study should be expanded or at least a motivation is necessary for the 3 interaction schemes that have been selected. What makes exactly those relevant?

Reviewer #1 indicate that he could not access the software, but this turned out a technical issue - I have asked him to have a second look and indeed he joins reviewer #2 who also writes that they had some issues getting the code up-and-running. They request that the README file is expanded and revised to assist users on getting the code up and running. The specific comment  by reviewer #1 was "Thank you for figuring that out. I believe I tried finding the repository via Google at the time, but perhaps it wasn't yet indexed because it is now easy to find. I have taken a look at the repository quickly. As I do not have the time to test the software now, I will only advise the authors to write a proper README file to elaborate on the usage of the software." 

Finally, please note that the comments of reviewer #2 have been uploaded as an attachment.

We cannot make any decision about publication until we have seen the revised manuscript and your response to the reviewers' comments. Your revised manuscript is also likely to be sent to reviewers for further evaluation.

Sincerely,

Roeland M.H. Merks, Ph.D

Academic Editor

PLOS Computational Biology

Kiran Patil

Section Editor

PLOS Computational Biology

Reviewer's Responses to Questions

**Comments to the Authors:**

Reviewer #1: In this paper, the authors study how cell-cell interactions and cell-environment interactions shape microbial population structuring. They use an individual-based model of three species (B1, B2, and B3), which are identical in kinetic parameters, but differ in their metabolic stoichiometry. The authors unpack the impact of three interaction regimes: neutral [0,0,0], competitive [-,-,-], and commensal [0,+,+]. They focus in particular on two modes of stratification, namely layered and columned stratification. They show that when resources are not the limiting factor (at high nutrient concentration), no stratification occurs, and microbial species simply form spatially separated clusters in space (note, as per my comments below, I would not call this well-mixed!). When resources are limited, a competitive environment results in columned stratification. However, when commensal interactions dominate, layered (concentric) patterns emerge. They finally derive a theoretical equation that allows them to differentiate between conditions that deliver either columned or layered stratification.

Firstly, I would like to congratulate the authors on this generally well-written paper filled with intriguing results. As a colour-blind person, I also totally appreciate the friendly colours that were chosen for this work. Especially figure 3C and others that show branching due to diffusion-limited aggregation are astonishingly beautiful. I am very supportive of publishing this work, but do suggest a revision round to polish this paper. There are a couple of typos and awkward sentences found throughout the manuscript, and there are a few major points I would like to see addressed. Glancing at my large list of major/minor points, I feel like it suggests I did not enjoy the work, but quite the contrary. I am very invested making sure this work is presented exactly as exciting as it is.

Major points:

1. I would encourage the authors to better explain the difference between the kinetic growth parameters (assumed to be identical) and the stochiometric constrains (assumed to be variable). At multiple places in the text, I got confused at to what exactly was identical, and what was different. I had to go back to the supplement multiple times, and that is not ideal for your readers. A suggestion would be to design a little cartoon of the stoichiometry to complement the text-based descriptions (e.g. [0,+,+]), which you can then add above each column in the figures so that the readers immediately know what that implies.

I also find it surprising that these three main interaction regimes that are studied are not mentioned in the abstract specifically.

Finally, just out of curiosity, antagonism wasn’t studied. Why not? I would love to see how your framework responds to antagonistic interactions. I would expect stratification (either concentric or columned) to perhaps break completely under many conditions. I understand this is clearly a case of “beyond the scope of this paper”, but perhaps explain at least why it is missing?

2. The results / discussion are dominated by results, and very thin on discussion. I would love to put these results in light of more data (interactions ranges by Alma Dal Co, mutational dynamics by Diane Fusco), and to also elaborate a bit more on how these colony dynamics (mostly derived from lab results) could translate to microbes leaving on a grain of soil, where there is much more lysis through phage predation (whereas your model, doesn’t take into account lysis at all). Plus there’s a variety of other stressors in nature that a lab cannot possibly capture (nor should they try to, this is what models like yours are for!). This is not to say I would encourage to change the model, but just to make it very clearer what you did *not* do. This will inspire ideas for future work, which I think is important in discussion sections.

3. I find the link with Sara Mitri’s work interesting, but it requires much more unpacking. Neither 100mM or 10mM give a “well-mixed” result, but I suppose your main point is to say they are not stratified. Acknowledging this may actually make the result more, not less interesting. Mitri really shows “well-mixed” outcomes, while your simulations still show something I would call far from “randomly distributed”. In Mitri’s work, resources are assumed to be truly mixed (if I remember correctly). Have you tried doing to same to confirm that particular result? If so, it would be a great addition to the supplement. I do appreciate the key point here, in that there is no difference between the 3 interaction modes, but I would really suggest to not call this “randomly distributed”, as they still clearly form spatially separated groups.

4. I really enjoy the final results (Fig 6 and 7), but I am wondering to what extend you could explain this without taking O2 into consideration. Can we see layered vs. columned stratification when *not* changing between two limiting resources (O2 and resource A), but simply modifying other parameters? Looking at equation 1 (for the eco-interaction modulus), I feel like the same differentiation of layered vs. columned stratification can be found by playing with the diffusion constant (D) and the concentration (C), giving either a phi_ei greater than 1, or smaller than 1. Perhaps the rates you’d have to assume for this are unreasonable biologically, but in that case that is also very interesting to mention at in the text, as that would imply that these differential patterns cannot be explained by a single limiting nutrient.

5. The concept of the dynamical fitness landscape is introduced in the introduction, but there is no further reference to it. Although I can see that your model fits this concept, it is not actually adding anything to this theory explicitly. Similar to point 2, perhaps discuss it some more later, or, not mention it in the introduction at all.

6. The GitHub link in the methods was broken, so I couldn’t actually confirm the published code is presented clearly. As someone who spends a lot of time trying to install tools, I would appreciate it if I can have a look to confirm everything is in order.

Minor points/typos:

1. This is subjective, but I find “eco-interactions” to be an unnecessary abbreviation of ecological interactions.

2. L37 Typo (reveals)

3. L69 Typo (ammonia-oxidizers)

4. L69 I think I have some more issues with that sentence. I know what is said, but I find it hard to parse. I’m not even sure it is grammatically correct (?).

5. L89 Our results suggests that ecological environment would control spatial organization -> no surprise. What is interesting about your model is not *that* is shapes spatial patterns, but *how* it shapes is.

6. L124 Protection only considers two species?

7. L296 Not sure if the word “even” in this sentence is necessary or clarifying at all.

8. L235 I had to dive into the stoichiometry to figure out why B2 and B3 have different growth dynamics. I think you may want to introduce here that B2 grows on B1’s products, and B3 on B2’s products, because as you phrase it now there is too much stress on “how the kinetics are identical”, which makes this result confusing for the reader. (this was one of my major points, but this sentence here is an example where the confusion arises)

9. L337 The word concentric is mentioned here, but nowhere else in the text. For me it actually would have helped if you coined this earlier, because I had to keep looking back at the figures to understand layered vs. columned stratification, while the word “concentric” would clear that right up.

10. L340 This sentence makes it sound like nobody believes the environment controls these patterns at all. I would rather read “[…] the environment has a significant impact on spatial distribution patterns as observed in nature.”, or something along those lines.

11. Figure 3B It’s unclear what the three statistical groupings are. The caption explains this, but I would suggest adding a line as is common practice.

12. Figure 4 -> Same layout as three, so same remarks.

13. L370 I wasn’t immediately sure what R(x,y,t) did, only when I got to line 379. The introducing sentence may need further unpacking (“A term of reaction is added function…”), as the grammar is also awkward.

14. L379 Alright, so I understood now. You may want to add to this section that you are assuming steady-state for the production/degradation, etc. � You do mention this later in L421, but I find it may help to mention this also here.

15. L390 Please make explicit that you assume no stochastic death (only through shrinking) and assume no lysis of cellular “products”.

16. L446 Is the word “the relationship” missing in the last part of this sentence? Otherwise, not sure what this sentence means.

17. L449 For clarity, is the initial distribution the only source of stochasticity in the model, or is there noise on other processes?

18. L476 Please explain the variables in the order they appear in equation 11-13, this is very confusing.

19. SupS1 Check grammar on page 12, “Due to bacteria will not move very far from original position, it is assumed that chosen neighbouring bacteria stay constant”

Reviewer #2: The review is uploaded as an attachment

Reviewer #3: This manuscript examines the spatial organization that emerges from interspecies interactions using an individual-based model. The main contribution of this manuscript, in my opinion, is the expansion of previous work to investigate more species (more than two) and multiple, concurrent interactions. I especially found the discussions around nutrient availability informative. I do not have any major comments or concerns, but I think addressing several minor issues listed below would improve the manuscript.

Minor comments:

1. Figure 1: Only three out of the six possible ecological interactions are studied in this manuscript (even without counting the many more combinations arising from concurrent interactions). I think it is fine to focus only on the three cases discussed in this manuscript; however, it would be helpful to include either a justification of why these three are chosen (perhaps in the introduction), or a brief description/speculation of what happens in the other cases (perhaps in the discussions).

2. Line 169: Please define “mass transfer resistance” in this context

3. Figure 2: I think it would be helpful to show examples of the initial distribution of cells (i.e. the inoculum) in the figure as well.

4. There are existing examples of three-species spatial patterning (simulations and experiments) in (Breugelmans et al., FEMS Microbiol Ecol, 2008, DOI:10.1111/j.1574-6941.2008.00470.x), and (Momeni et al., eLife, 2013, DOI:10.7554/eLife.00960), which I think are in line with authors’ findings and can be mentioned as supporting evidence. I think, an expansion of the contributions in prior work (many of them cited in the introduction) would have been nice (optional).

5. Related to Figure 2, the “layered stratification” as described in the text is not immediately obvious when visually examining the patterns in Fig 2. This is perhaps also related to the previous comment in which existing examples more clearly exhibit “columned stratification” and “layered stratification”, when the cross-sections of communities are examined. While I understand that the authors in the context of microbial aggregates have focused on the (x,y) patterns in this manuscript, perhaps the terminology can be reconciled and justified by mentioning the relation with the (x,z) coordinates.

6. Line 247: I would suggest not using the term “evolution” to describe changes in community properties during the emergence of spatial organization to avoid confusion.

7. Lines 396, 632, and 634: Again the term “evolution” is used in a context that might cause confusion

8. I don’t think the growth rate comparisons at different resource levels in Fig 5D are fair, since the amount of growth (say, quantified as the number of generations of total population growth) is very different at different [S]_T levels

9. The link to the codes should be: https://github.com/Computational-Platform-IbM/IbM (it's written correctly, but the hyperlink points to the wrong address).

**Have the authors made all data and (if applicable) computational code underlying the findings in their manuscript fully available?**

Reviewer #1: **No: **Link was broken and a quick google search didn't get me anywhere either.

Reviewer #2: Yes

Reviewer #3: Yes

PLOS authors have the option to publish the peer review history of their article (what does this mean?). If published, this will include your full peer review and any attached files.

Reviewer #1: **Yes: **Bram van Dijk

Reviewer #2: No

Reviewer #3: No
---

## [Decision Letter · Decision Letter 1]

23 Nov 2022

Dear Dr. Martinez-Rabert,

Thank you very much for submitting your manuscript "Environmental and ecological controls of the spatial distribution of microbial populations in aggregates" for consideration at PLOS Computational Biology. As with all papers reviewed by the journal, your manuscript was reviewed by members of the editorial board and by several independent reviewers. The reviewers appreciated the attention to an important topic. Based on the reviews, we are likely to accept this manuscript for publication, providing that you modify the manuscript according to the review recommendations.

There are still some key issues regarding the assumption of independence of the colony sizes that may affect the statistics, and may therefore your conclusions. For example, as pointed out by reviewer #2, in Figure 4 you argue that the colony sizes can be independent, because substrate levels are constant, and therefore you argue the colonies may be considered independent. However, since bacteria cannot accumulate, a strong competition for space will remain and, therefore, the colonies are dependent of one another. In your revised manuscript, please address these statistical issues and the other, smaller issues addressed by reviewer #2 and the Github issue raised by reviewer #3.

Sincerely,

Roeland M.H. Merks, Ph.D

Academic Editor

PLOS Computational Biology

Kiran Patil

Section Editor

PLOS Computational Biology

Reviewer's Responses to Questions

**Comments to the Authors:**

Reviewer #1: Dear authors,

You went above and beyond with this revision. I have recommended this manuscript for publication.

Reviewer #2: Review of "Environmental and ecological controls of the spatial distribution of microbial populations in aggregates"

The paper is a resubmission, and many important changes and additions have been made, which greatly improve the paper. The authors now present their model of ecological interactions in bacterial aggregates in a clearer way, and include more comparison to the existing literature. Extra analysis of a shearing condition has been added, which I think is a great additional topic to be studied with this model. The clearer documentation of how to run the model is also very welcome. The technique of individual-based computational modelling to represent bacterial aggregates remains interesting, as does the eco-interaction modulus. Overall, the authors have written a fine paper and use a novel method to produce interesting results. However, I still have a number of remarks on various aspects of the paper that I think should be adressed.

Major remarks

1. I do not understand your explanation for treating each microbial colony as an independent observation in fig 4D. These differences in colony size must be a result of the (indirect) interactions with other colonies. Treating them as independent overestimates your number of truly independent observations. I would expect each run to produce a single number (mean colony size), and for these to be compared, or for some other statistical method to be used that takes into account that observations are not independent.

2. Similarly, I am missing an argument for the independence of observations in figures such as 3B. I would instead argue that the population of one species cannot be considered independent from that of another species in the same simulation, due to competition for space, and so cannot be compared using this test. A paired t-test would be more appropriate for this type of observations. If results remain significant for the neutralism condition, some explanation would be good on why there are three significant pairwise interactions out of 9. It may be be purely stochastic variation, of course, but I would urge you to make sure the initial seeds are different, and there is no bug in your code leading to species 2 and 3 being favored.

(I would not expect the different statistical methods to change the significance of any other results - they seem quite clearly different to me from the plots)

3. The statistics performed in figure 5 are unclear to me. What is the meaning of the width of the lines under the stars? And what, exactly, is being compared there? Intuitively, I would expect the two lines to be compared, but you say in the caption that you are comparing the 1mM condition with the other conditions, which are all combined into a single line. It would be much clearer to represent each condition (or maybe only the 1mM and 0.1mM, for clarity) as its own set of two lines, especially if there are significant differences between them.

4. I do not understand your conclusions on shear. Looking at S3Video (the figure is compressed a bit), it seems to me that layered stratification is still present in the commensalism and the commensalism+competition condition. While not as clear as in the no detachment condition, the average location of B3 populations seems closer to the centre than that of B2, which is closer than that of B1. This is of course only based on a single picture - your hypothesis may very well still be correct. It would be good to have a few runs and some statistical analysis of this behaviour - it may also depend on the initial configuration.

5. The conclusion at lines 419-420 and 430-433 is stated too strongly - your results do not show that space plays a fundamental role in in vivo or in vitro microbial aggregates.

6. I do not understand your conclusion that neutralism is a transient state (line 422). It implies that you do not consider your neutralism condition to be neutralism, as it does feature a competition for space. If this is the case, a different term should be used for ‘true’ neutralism, and neutralism with competition for space, to prevent confusion.

Minor remarks

1.Your instructions to run the code are incorrect on a crucial point: 7.II says the user should run the code with “IbM(sim_xxxx)”, but the user should run “IbM(xxxx)”.

2. S1Fig is a good addition, but you should define the phrase “diffusion resistance”, as used in the caption, or use a different phrase. Is the model simply well-mixed if this is disabled? In addition, it is unclear why the inoculum size is also changed between the two conditions, and whether these are at steady state or not.

3. Line 94: You state that neutralism, competition, and commensalism are the most common bacterial interactions, but the citation (“Bacterial species rarely work together”) shows that, generally, it is competition, exploitation and amensalism that are the most common interactions.

4. At line 122-124 you still say that all simulations are performed until a steady state is reached, but this is not the case for the first results you show (Fig. 2)

5. The visualisation with boxplots in, e.g., 3B is not ideal when n=3 or n=6 - plotting the individual data points would be more informative with such a low n.

6. line 176: you refer to Fig 4A, but I think you intend to refer to 3A

7. The p-values in fig S3B do not match what you report at line 218

8. Line 429 and table S3: you still mention protective populations here, but nowhere else. In addition, the supplemental figures referred to in table S3 are misnumbered.

9. Line 508: what's the difference between a minimum mass and a negligible size? Also, what does it mean for a cell to be inactive? I assume it would not grow, but you mention in line 509 that it can become active when it increases in mass. How can this happen?

10. Scale bars are missing units in Fig. 8

11. How are random seeds determined? Line 542 implies that each simulation gets a different random seed, but in S3_Video the top three simulations have the same initial condition as the bottom three simulations. In addition, the section from line 538-543 claims all simulations are run in triplicate, but some are run six times.

12. Editing for language is still needed - many lines have grammatical errors in them. Some examples:

70 "are collaborating" should be "collaborate"

101 "how ecological" missing "the"

102 "control" should be "controls"

153 There should be no comma here

166 "are" should be "is", or "structure" should be "structures"

284 “as” -> “such as”

302 “when concentration” - > “when the concentration”

398-399 “the same trend than Suarez et al al was observed”

422 “for” -> “of”

473-474 “one” (singular) and “are” do not match

478 "kinetics function" - should be "kinetics as a function", I think

504 "remained" should be "remaining"

550-551 This sentence is very hard to follow

576 "it was defined a"

739 no -> not

Reviewer #3: In my opinion, the authors have adequately addressed all reviewers' comments. The only remaining issue is that the link to the GitHub repository is still broken, but that might have happened during the pdf conversion. This can be fixed in the final production.

**Have the authors made all data and (if applicable) computational code underlying the findings in their manuscript fully available?**

Reviewer #1: Yes

Reviewer #2: **No: **The data points behind means etc. do not seem to be available. They are not mentioned in the data availability statement, and I cannot find them on the Github listed. My apologies for not noticing this in the first round of review.

Reviewer #3: Yes

PLOS authors have the option to publish the peer review history of their article (what does this mean?). If published, this will include your full peer review and any attached files.

Reviewer #1: **Yes: **Bram van Dijk

Reviewer #2: No

Reviewer #3: No

Figure Files:

Data Requirements:

Reproducibility:

References:

---

## [Decision Letter · Decision Letter 2]

10 Dec 2022

Dear Dr. Martinez-Rabert,

We are pleased to inform you that your manuscript 'Environmental and ecological controls of the spatial distribution of microbial populations in aggregates' has been provisionally accepted for publication in PLOS Computational Biology.

Best regards,

Roeland M.H. Merks, Ph.D

Academic Editor

PLOS Computational Biology

Kiran Patil

Section Editor

PLOS Computational Biology

Reviewer's Responses to Questions

**Comments to the Authors:**

Reviewer #2: My comments have been addressed very well, I am satisfied with the paper in this form.

I only have one small grammar mistake to note: on line 400-401 “like in Suarez et al. study” should be “like in the study by Suarez et al.”

**Have the authors made all data and (if applicable) computational code underlying the findings in their manuscript fully available?**

Reviewer #2: Yes

PLOS authors have the option to publish the peer review history of their article (what does this mean?). If published, this will include your full peer review and any attached files.

Reviewer #2: No

---

## [Editor Report · Acceptance letter]

15 Dec 2022

PCOMPBIOL-D-22-01033R2 

Environmental and ecological controls of the spatial distribution of microbial populations in aggregates

Dear Dr Martinez-Rabert,

I am pleased to inform you that your manuscript has been formally accepted for publication in PLOS Computational Biology. Your manuscript is now with our production department and you will be notified of the publication date in due course.

With kind regards,

Zsofia Freund
